Retrospective study: clinicopathological features and prognosis of idiopathic membranous nephropathy with seronegative anti-phospholipase A2 receptor antibody

Guo Wenkai 1 2
Zhang Yan 3
Gao Caifeng 4
Huang Jing 4
Li Jiatong 1 5
Wang Rong 1
Chen Bing chenbing3668@163.com 1
1 Department of Nephrology, Shandong Provincial Hospital Affiliated to Shandong University , Jinan , Shandong , China
2 Department of Nephrology, Shandong Provincial Hospital Affiliated to Shandong First Medical University & Shandong Academy of Medical Sciences , Jinan , Shandong , China
3 Department of Nephrology, Yantai Yuhuangding Hospital, School of Medicine, Qingdao University , Yantai , Shandong , China
4 Department of Nephrology, Jinan Shizhong People’s Hospital , Jinan , Shandong , China
5 Department of Geriatrics, Shandong Provincial Hospital Affiliated to Shandong First Medical University & Shandong Academy of Medical Sciences , Jinan , Shandong , China
Kim Cheorl-Ho
Electronic publication date: 2020 Feb 21
Publication date: 2020
Volume: 8
Electronic Location ID: e8650
Received 2019 Sep 5; Accepted 2020 Jan 28
Copyright: ©2020 Guo et al.
Copyright year: 2020
Copyright holder: Guo et al.
License: This is an open access article distributed under the terms of the Creative Commons Attribution License, which permits unrestricted use, distribution, reproduction and adaptation in any medium and for any purpose provided that it is properly attributed. For attribution, the original author(s), title, publication source (PeerJ) and either DOI or URL of the article must be cited.
License URL: https://creativecommons.org/licenses/by/4.0/

Keywords: Serum phospholipase A2 receptor, Idiopathic membranous nephropathy, Cyclophosphamide, Calcineurin inhibitor, Remission rate

Funding: Primary Research & Development Plan of Shandong Province 2018GSF118227 Science and Technology Plan This study was supported by grants from the Primary Research & Development Plan of Shandong Province (2018GSF118227) and the Science and Technology Plan (673 and 741) of Shizhong District of Jinan City. The funders had no role in study design, data collection and analysis, decision to publish, or preparation of the manuscript.

==============================
Background

To discuss the clinicopathological features and prognosis of patients with idiopathic membranous nephropathy (IMN) who are serum-negative for the anti-PLA2R antibody.

Method

Overall, 229 IMN patients were retrospectively collected in this study and classified into anti-PLA2R antibody-negative (PLA2R−, 59 cases) and antibody-positive (PLA2R+, 170 cases) groups. The clinical and pathological features of the PLA2R− group were analyzed; 162 patients in both groups were followed up, and the PLA2R antigen was detected in renal biopsies from the PLA2R− group. Kaplan-Meier and survival analyses were used to compare differences in prognosis.

Results

Serum albumin levels were higher and 24-hour urine protein, creatinine, and beta 2-microglobulin (BMG) levels were lower in the PLA2R− group than in the PLA2R+ group; the proportion of acute and chronic tubular lesions was also significantly lower in the PLA2R− group than in in the PLA2R+ group. After treatment, the remission rate was significantly higher in the negative group than in the positive group (93.02% vs 74.78%,), especially the rate of complete remission (51.16% vs 23.47%). Furthermore, the PLA2R antigen-positive staining rate of 43 patients in the PLA2R− group was 62.79%. Although not significant, the survival rate was higher in the PLA2R− group than in the PLA2R+ group. BMG, 24-hour urine protein and acute and chronic tubular lesions were risk factors for kidney death, and 24-hour urine protein was an independent risk factor for kidney death.

Conclusions

Compared with the PLA2R+ group, the PLA2R− group had mild clinical manifestations and pathological damage and a higher clinical treatment remission rate. Renal tissue PLA2R antigen testing can be considered for patients with seronegative IMN to increase the diagnostic rate.

Introduction

Membranous nephropathy with unclear etiology is called idiopathic membranous nephropathy (IMN) and accounts for 30∼40% of all cases of primary nephrotic syndrome (Braden et al., 2000; Couser, 2017). Recent clinical observations show that the incidence of membranous nephropathy is increasing, and the age of onset is becoming younger, which may be related to various factors such as improvements in diagnostic factors and environmental pollution (Ayach et al., 2011). Moreover, IMN is also a heterogeneous disease with a long natural course. Approximately one-third of patients will exhibit variable degrees of persistent proteinuria without deterioration in renal function, another third will experience progressive deterioration of renal function and eventually progress to end-stage renal disease (ESRD), and the final third of patients will experience spontaneous remission (Obrisca et al., 2015).

Currently, the method for diagnosing membranous nephropathy is mainly based on renal biopsy, an invasive examination method that has inherent risks and cannot be used to diagnose and treat diseases in a timely and effective manner. Therefore, the serological diagnosis of IMN has become a hot topic in kidney disease research. Beck et al. (2009) discovered autoantibodies against phospholipase A2 receptor antibodies in the glomerular podocytes of IMN patients. PLA2R is a transmembrane protein located on the surface of human podocytes and colocalizes with IgG4 in the immune deposits of glomeruli in patients with IMN. Binding of the circulating anti-PLA2R antibody to the PLA2R antigen on glomerular podocytes to form an in situ immune complex activates a complement to cause podocyte and immune damage that results in urinary protein production, ultimately causing kidney damage (Glassock, 2012). Additionally, Beck et al. found that serum anti-PLA2R antibodies were detectable in 70% of patients with IMN and detected at a low rate in people without kidney disease and other kidney patients (Hofstra & Wetzels, 2014). Further studies have found that the antibody level is related to the urine protein level, disease condition, and prognosis and has high sensitivity and specificity in the clinic (Hofstra et al., 2011; Hoxha et al., 2011; Hoxha et al., 2014). In addition to serum PLA2R antibodies, a number of clinical studies (Beck et al., 2009; Dai, Zhang & He, 2015; Svobodova et al., 2013) have found that the detection of PLA2R antigen in renal tissue also plays an important role in the clinical diagnosis, guiding treatment and judgment of disease activity of IMN. The expression of PLA2R antigen in renal tissue was significantly enhanced in IMN patients, and low or no expression was observed in patients with secondary membranous nephropathy (SMN) and other nonmembranous glomerular diseases, indicating that the PLA2R antigen in renal tissue also has high specificity in the diagnosis of IMN.

Several studies have shown that the anti-PLA2R antibody titer reflects the activity of the clinical disease. The low antibody titer in antibody-positive patients indicates a high disease remission rate, and thus, this antibody can be used as a predictor of clinical efficacy (Hofstra et al., 2012; Oh et al., 2013). However, there are few studies on the clinical and pathological characteristics of patients with IMN who are negative for the serum anti-PLA2R antibody, and the effect of a negative antibody on disease prognosis is unknown. Besides, some studies suggest that patients with negative antibodies may have secondary factors, such as tumors and heavy metals (Radice et al., 2018). Therefore, the characteristics of serum anti-PLA2R antibody-negative IMN require further clinical studies.

In this study, we performed a two-step evaluation. First, we retrospectively compared the clinical and pathological features of anti-PLA2R−negative (PLA2R−) and anti-PLA2R−positive (PLA2R+) subjects, and the PLA2R antigen was detected in renal biopsies from the PLA2R− group. Then, we evaluated the prognostic value of PLA2R−negative results using a longitudinal design and analysis.

Materials and Methods

Patient selection

From January 2015 to December 2017, 229 adult patients at Shandong Provincial Hospital who were diagnosed with nephrotic syndrome confirmed by renal biopsy as IMN and had complete clinical data were included in this study. All patients had IMN, a condition that is universally diagnosed by kidney biopsy based on the presence of subepithelial spikes along capillary walls as determined by silver staining, granular IgG and C3 along capillary walls as determined by immunofluorescence, and subepithelial deposits as determined by electron microscopy (EM). The clinical study excluded patients with SMN caused by autoimmune disease, tumors, and hepatitis B virus- and metabolic-related diseases. At the time of selection, the patients’ renal function was in the normal range, and 162 patients were followed up for more than 6 months with complete follow-up data. This study has obtained verbal consent from patients, as well as reviewed and approved by the Medical Ethics Committee of Shandong Provincial Hospital (LCYJ: no. 2019-105).

Clinical and laboratory data collection

General demographics (age, gender) and clinical (duration of disease, systolic and diastolic blood pressure, etc.) records were collected. Laboratory data included general tests (urinary red blood cells, hemoglobin, white blood cells, platelets, alanine aminotransferase (ALT), aspartate aminotransferase (AST), superoxide dismutase (SOD), serum total protein (TP), albumin (ALB), globulin (GLO), cystatin C (Cys-C), retinol-binding protein (RBP), β2 microglobulin (BMG), complement C1q, calcium, immunoglobulin, total cholesterol (CHOL), low-density lipoprotein (LDL-C), high-density lipoprotein (HDL-C), antibody, antinuclear antibody, ANCA antibody, blood glucose (GLU), etc.) and specific biomarkers of renal function and disease activity (urine protein, 24-hour urine protein, serum creatinine (CREA), and blood urea nitrogen (BUN)). The eGFR was calculated based on the modification of diet in renal disease (MDRD) formula.

Anti-PLA2R detection method

The anti-phospholipase A2 receptor antibody IgG detection kit (enzyme-linked immunosorbent assay) produced by EUROIMMUN (product no. EA 1254-9601 G) was used and performed according to the standard procedure in the manual using a normal reference range of 0–20 RU/ml. The study also included antibody-positive components in the low-titer group (20–180 RU/mL) and high-titer group (>180 RU/mL) (Hofstra et al., 2012).

PLA2R antigen and IgG4 detection

We collected paraffin sections of kidney biopsies from 43 patients negative for anti-PLA2R antibodies, used direct immunofluorescence to detect the deposition of IgG4 in renal tissue and indirectly detected the PLA2R antigen in renal tissue. The fluorescence intensity was semiquantitatively recorded by fluorescence microscopy and scored as negative (−), suspicious positive (±), and positive (+).

Pathological data

Each patient’s renal biopsy included at least 10 glomeruli for histopathological evaluation. We performed light microscopy, electron microscopy (EM) and immunofluorescence on renal puncture tissues. Two renal pathologists participated in the reading and review of the pathological results. We divided membranous nephropathy into 4 stages. If two stages were present at the same time, we defined the highest stage as the final stage and stages III and IV as the advanced pathological stages. Simultaneously, this study observed the presence or absence of focal segmental glomerular sclerosis, spheroidal sclerosis, mesangial proliferative lesions, crescent incidence, balloon adhesions, endothelial hyperplasia, acute and chronic tubular lesions, inflammatory cell-infiltrating lesions, small blood vessel lesions, etc. The pathological manifestations of glomerular disease combined with focal segmental glomerulosclerosis (FSGS) showed focal and segmental distribution of glomerular lesions under light microscopy, with an increased mesangial matrix and balloon adhesions as the main manifestations, accompanied by a small amount of mesangial hyperplasia, corresponding tubular atrophy and renal interstitium fibrosis. Spherical sclerosis is defined based on the hardening of one glomerulus, and crescent incidence is defined as the presence of a cellular or fibrous crescent in one glomerulus. Mesangial proliferative disease is defined as the number of cells in the mesangial area being greater than or equal to 4. Acute tubulointerstitial lesions are defined as tubular epithelial edema or necrosis of at least 10%. Chronic tubulointerstitial lesions are defined as astubular atrophy and interstitial fibrosis of at least 10%. Interstitial inflammatory disease mainly manifests as neutrophil infiltration. Small vessel lesions include arterial intimal thickening, elastic layer stratification, and hyaline degeneration.

Outcomes in followed up patients

We defined worsening renal conditions as doubling of the baseline Scr level; we defined ESRD as a creatinine clearance rate of less than 15 ml/min at the last follow-up, start of dialysis or renal transplantation. Complete remission (CR) was defined as a urine protein level <0.3 g/d based on the premise of stable renal function. Partial remission (PR) was defined as a 50% reduction in the urine protein to a proteinuria level <3.5 g/d. A simplified MDRD formula was used to calculate eGFR as follows: eGFR [ml min−1 (1.73 m2)−1] = 186 × [Scr (µmol/L)/88.4]−1.154 × Age−0.203 × (if female, 0.742). Serious complications included clinical death, serious lung infection, pulmonary embolism, cerebral infarction, myocardial infarction, and tumors. All patients underwent eGFR measurements at the beginning and after treatment.

Statistical methods

Statistical analysis was performed using the statistical software SPSS 19.0. Data with a normal distribution were expressed as mean  ± standard deviation (SD) and compared by t-tests, and data with a non-normal distribution were presented as the median and quartile and compared by nonparametric test. The categorical variables were expressed as rates and were compared by the χ2 test. Correlation between several parameters was expressed as Spearman rank coefficient of correlation. Patient and renal survival probabilities were estimated by the Kaplan- Meier method. The relationships of the covariates to patient and renal survival were evaluated in the univariate analysis with the log-rank test and in the multivariate analysis with the Cox proportional hazards model; the Harrel C statistic was used for verification of the risk factors confirmed by the Cox proportional hazards model. A Cox analysis of risk factors was performed in combination with clinical significance. P < 0.05 was considered statistically significant.

Table 1 Comparison of clinical characteristics and basic laboratory tests between the two groups.

	IMN (N = 229)	APLA2R-ab(-)(N = 59)	APLA2R-ab +(N = 170)	P value	
Age (year)	45.88 ± 13.64	45.75 ± 13.49	45.92 ± 13.73	0.931	
Sex (female (%))	91/229 (39.74%)	30/59 (50.84%)	61/170 (35.88%)	0.043*	
Duration of illness (day)	117.72 ± 175.40	129.57 ± 210.58	112.44 ± 160.84	0.542	
SBP (mmHg)	137.70 ± 20.48	135.63 ± 23.30	140.90 ± 21.03	0.190	
DBP (mmHg)	87.10 ± 13.89	88.93 ± 16.66	88.08 ± 13.93	0.626	
Urine RBC/HPF	14.07 ± 17.76	11.24 ± 14.87	15.04 ± 18.59	0.160	
24HUPRO (g/L)	6.12 ± 8.17	4.91 ± 3.18	5.81 ± 3.50	0.086	
WBC (×109/L)	8.28 ± 17.76	11.95 ± 34.47	7.00 ± 2.81	0.275	
HGB (g/L)	134.28 ± 30.47	132.18 ± 38.80	135.01 ± 27.05	0.605	
PLT (×109/L)	268.09 ± 71.73	270.17 ± 95.40	267.36 ± 61.64	0.833	
AST (u/L)	22.67 ± 8.20	23.26 ± 8.61	22.46 ± 8.07	0.525	
ALT (u/L)	22.83 ± 11.96	25.37 ± 12.18	21.96 ± 11.79	0.063	
SOD (u/mL)	111.83 ± 27.18	115.83 ± 28.70	110.41 ± 26.59	0.245	
TP (g/L)	47.53 ± 7.60	49.17 ± 8.28	46.97 ± 7.30	0.056	
ALB (g/L)	24.69 ± 5.42	26.11 ± 6.02	24.20 ± 5.11	0.019*	
GLO (g/L)	22.92 ± 4.61	23.62 ± 4.67	22.68 ± 4.58	0.183	
GLU (mmol/L)	5.36 ± 1.30	5.59 ± 1.99	5.28 ± 0.93	0.248	
BUN (mmol/L)	5.36 ± 1.92	5.44 ± 1.93	5.33 ± 1.93	0.702	
CREA (μmol/L)	64.78 ± 15.78	61.57 ± 15.88	65.90 ± 15.64	0.033*	
Cys-C(mg/L)	0.99 ± 0.48	1.06 ± 0.81	0.97 ± 0.28	0.427	
BMG (mg/L)	2.32 ± 0.80	2.13 ± 0.74	2.39 ± 0.81	0.045*	
RBP (mg/L)	55.97 ± 18.24	54.36 ± 16.44	56.51 ± 18.83	0.451	
Ca (mmol/L)	2.17 ± 0.21	2.17 ± 0.21	2.17 ± 0.22	0.972	
CHOL (mmol/L)	8.84 ± 2.68	8.62 ± 2.68	8.92 ± 2.69	0.485	
HDL-C (mmol/L)	1.66 ± 0.51	1.71 ± 0.60	1.65 ± 0.48	0.454	
LDL-C (mmol/L)	5.45 ± 2.25	5.14 ± 1.95	5.56 ± 2.34	0.243	
FSGS	41/229 (17.9%)	9/59(15.25%)	32/170 (18.82%)	0.538	
IgG4 (mg/L)	352.49 ± 317.78	305.83 ± 268.24	368.56 ± 332.43	0.220	
IgG (g/L)	5.74 ± 2.51	5.71 ± 2.27	5.74 ± 2.59	0.942	
IgM (g/L)	1.13 ± 0.51	1.17 ± 0.53	1.11 ± 0.50	0.449	
IgA (g/L)	2.34 ± 0.91	2.24 ± 0.86	2.37 ± 0. 86	0.360	
C3 (g/L)	1.19 ± 0.23	1.19 ± 0.18	1.18 ± 0.24	0.937	
C4 (g/L)	0.30 ± 0.08	0.28 ± 0.07	0.30 ± 0.18	0.112	
C1q (mg/L)	216.70 ± 38.59	213.19 ± 34.61	217.99 ± 40.01	0.463	
Notes.

SBP systolic blood pressure

DBP diastolic blood pressure

24HUPRO 24-hour urine

WBC white blood cells

PLT platelets

HGB hemoglobin

AST alanine aminotransferase

SOD superoxide dismutase

TP serum total protein

ALB albumin

GLO globulin

GLU blood glucose

BUN blood urea nitrogen

CREA serum creatinine

Cys-C cystatin C

BMG β2 macroglobulin

RBP retinol binding protein

CHOL total cholesterol

HDL-C high-density lipoprotein

LDL-C low-density lipoprotein

FSGS focal glomerular sclerosis

IgG4 immunoglobulin G4

IgG immunoglobulin G

IgM immunoglobulin M

IgA immunoglobulin A

C3 C4 complement 3 complement 4

PLA2R phospholipase A2 receptors

Compared with the PLa2R− group and the PLA2R+ group, P∗ < 0.05, P∗∗ < 0.01.

Results

General clinical data

Two hundred twenty-nine patients with IMN were retrospectively collected and analyzed in this study, including 59 (25.76%) patients in the PLA2R− group and 170 (74.24%) patients in the PLA2R + group. Among the antibody-positive group, there were 109 (64.12%) patients in the low-titer group and 61 (35.88%) patients in the high-titer group. The comparisons of baseline clinical characteristics and laboratory examination data between the two groups are shown in Table 1. In this study, the levels of serum creatinine (61.57 ± 15.88 vs 65.90 ± 15.64, df = 226, P = 0.033) and β2-microglobulin (2.13 ± 0.74 vs 2.39 ± 0.81, df = 209, P = 0.045) in the PLA2R− group were lower than those in the PLA2R+ group, and the level of ALB (26.11 ± 6.02 vs 24.20 ± 5.11, df = 226, P = 0.019) in the PLA2R− group was higher than that in the PLA2R+ group. The average 24-hour urine protein quantification in the PLA2R− group was slightly lower than that in the PLA2R+ group (4.91 ± 3.18 vs 5.81 ± 3.50), but the difference was not statistically significant (df = 224, P = 0.086). At the same time, we found no correlation between the anti-PLA2R antibody level and the serum creatinine level in the PLA2R+ patients (r =  − 0.08, P = 0.256). The anti-PLA2R antibody level was positively correlated with proteinuria (r = 0.151, P = 0.05) and negatively correlated with serum albumin (r =  − 0.164, P = 0.03). The 24-hour urine protein levels of patients in the low-titer group were significantly lower than those in patients in the high-titer group (5.28 ± 3.42 vs 6.76 ± 3.46, df = 166, P = 0.008).The eGFR values of the antibody-negative and antibody-positive groups were 115.394 and 109.860 [ml⋅min-1⋅ (1.73 m2)-1], respectively, but the difference was not statistically significant (df = 225, P = 0.172).

Pathological data

Renal biopsy immunofluorescence showed no significant differences in IgG, IgM, IgA, C3, Fib, or C1q deposits between the two groups. There was no significant difference in the proportion of renal pathological stages between the two groups (P > 0.05). The proportions of acute and chronic tubular lesions and inflammatory cell infiltration in the PLA2R− group were significantly lower than those in the positive group (respectively, 3.39% vs 11.76% df = 180, P = 0.015; 3.39% vs 13.53% df = 183, P = 0.01; and 5.08% vs 13.53% df = 158, P = 0.031) (Table 2).

Table 2 Comparison of pathological characteristics between the two groups.

	APLA2R-ab (−) (N = 59)	APLA2R-ab (+) (N = 170)	P value	
Pathological stage (%)	
I	45/59 (76.27%)	112/170 (65.88%)	0.454	
II	14/59 (23.72%)	55/170 (32.35%)	0.321	
III + IV	0	2/170 (1.18%)	0.293	
Tissue IgG	2.46 ± 0.83	2.37 ± 0. 89	0.486	
Tissue IgM	0.36 ± 0.79	0.38 ± 0.69	0.810	
Tissue IgA	0.12 ± 0.50	0.23 ± 0.61	0.182	
Tissue C3	1.09 ± 0.87	1.19 ± 0.94	0.459	
Tissue Fib	0.08 ± 0.39	0.15 ± 0.52	0.264	
Tissue C1q	0.53 ± 0.80	0.50 ± 0.71	0.796	
Pathological characteristics	
Global sclerosis	35/59 (59.32%)	88/170 (51.76%)	0.337	
Glomerular mesangial hyperplasia	16/59 (27.12%)	64/170 (37.65%)	0.124	
Crescents	2/59 (3.38%)	0	0.159	
Hyperplasia endothelialitis	4/59 (6.78%)	7/170 (4.12%)	0.418	
Acute renal tubular lesions	2/59 (3.39%)	20/170 (11.76%)	0.015*	
Chronic renal tubular lesions	2/59 (3.39%)	21/170 (13.53%)	0.010**	
Inflammatory cell infiltration	3/59 (5.08%)	23/170 (13.53%)	0.031*	
Vascular disease	21/59 (35.59%)	57/170 (33.53%)	0.796	
Notes.

Compared with the PLa2R− group and the PLA2R+ group, P∗ < 0.05, P∗∗ < 0.01.

Treatment and outcomes

Among the 229 patients, 162 patients, including 43 patients in the PLA2R− group and 119 patients in the PLA2R+ group, were followed up for more than 6 months, and the average follow-up time was 11.97 ± 3.84 months. We treated patients with individualized and specific treatments, including glucocorticoids plus immunosuppressants (cyclophosphamide (CTX), cyclosporine A (CSA), tacrolimus). Our treatment plans were as follows: patients on drug regimens were treated according to their blood pressure and blood lipids. Acetylcholinesterase inhibitors (ACEIs) and angiotensin type 1 receptor blockers (ARBs) were used to control blood pressure below 140/90 mmHg, and statins were used to control blood lipids. Depending on the presence of edema, diuretics were sometimes used. Tacrolimus was administered according to the following regimen: an initial oral dose of 0.5 mg/(kg d), with continued treatment for at least 6 months. The plasma concentration of tacrolimus was determined for 1 week to maintain of value of 5∼10 ng/ml. CSA was administered according to the following regimen: an initial oral dose of 3–5 mg/(kg d), with continued treatment for at least 6 months. The plasma concentration of CsA was monitored for 1 week to maintain a value of 100∼200 ng/ml. CTX was administered according to the following regimen: administration of a static dose of 750 mg/m2 body surface area for at least 6 months and a cumulative dose of 6–8 g. All patients were given a sufficient dose of prednisone 1 mg/(kg d). After 8 weeks of adequate treatment, the dose size was reduced by 5 mg every 2 weeks and then held constant at a low dose of 10 mg/d. The total course of treatment was at least 9 months. This study found that the overall remission rate (including CR and PR) in the PLA2R− group was 93.02%, which was significantly higher than that in the PLA2R+ group (74.78%), and the difference was statistically significant (χ2 = 6.474, df = 1, P = 0.01). The CR and PR rates in the negative group were 51.16% and 41.86%, respectively, and the CR and PR rates in the positive group were 24.37% and 50.42%. In addition, in the antibody-positive group, the low titer group had a higher remission rate than the antibody high titer group (81.58% VS 62.79%), and the difference was statistically significant (χ2 = 5.142, df = 1, P = 0.02). We further performed a subgroup analysis, which revealed that the CR rate of patients in the PLA2R− group was significantly higher than that of patients in the PLA2R+ group in the different treatment regimens (Table 3). Consistent with the above results, the serum albumin levels of both groups were increased, and the quantified 24-hour urine protein levels were decreased significantly after treatment. However, the levels of proteinuria (df = 154, P = 0.01), serum creatinine and urea nitrogen (df = 106, P = 0.03) in the negative group were lower than those in the positive group, and the albumin level (df = 156, P = 0.036) in the negative group was higher than that in the positive group (Table 4). After treatment, the eGFR values of the negative and positive groups were 107.77 ± 24.431 and 105.20 ± 28.09 [ml min-1 (1.73 m2)-1], respectively, but the difference was not statistically significant. Moreover, for patients with IMN with follow-up data, the reduction rate of the anti-pla2r antibody after treatment was 92%. In addition, we detected both the PLA2R antigen and IgG4 in the renal tissues of 43 patients with negative antibodies by staining, revealing that 27 (62.79%) people were positive for the PLA2R antigen, 29 (67.44%) people were positive for IgG4, and 25 (58.14%) people were positive for both (Fig. 1). By the end of the follow-up, a total of 5 patients had worsening renal conditions, and all 5 patients were in the PLA2R+ group. Four of the patients were in the hormone plus calcineurin inhibitor treatment group, and of these, 2 patients had excessively high blood drug concentrations, 2 patients had some relief of proteinuria, and 2 patients had no improvement of proteinuria. Another patient was assigned to the hormone plus CTX group, and the patient’s proteinuria did not turn negative. During the follow-up process, we found one case of a benign lung tumor and three cases of severe complications, including 2 cases of severe pneumonia (1 death without renal failure) and 1 case of intracranial fungal infection, in the PLA2R+ group (Table 5).

In this study, we used the Kaplan–Meier method to map the survival curve. The log-rank test showed no significant difference in renal survival between the two groups (χ2 = 1.586, P = 0.208) (Fig. 2). At the same time, we conducted risk factor analysis, revealing that BMG, 24-hour urine protein and acute and chronic tubular lesions were risk factors for kidney death. The Cox proportional hazard model analysis showed that 24-hour urine protein was an independent risk factor for renal death (P = 0.034) (Tables 6 and 7).

Table 3 Comparison of remission rates after 1 year of follow-up and treatment in two groups ofpatients.

Total	Complete remission rate (CR)	Partial remission rate (PR)	CR+PR	
MN (N = 162)	51/162(31.48%)	78/1629 (48.15%)	129/162 (79.63%)	
PLA2R − (N = 43)	22/43 (51.16%)	18/43 (41.86%)	40/43 (93.02%)*	
PLA2R + (N = 119)	29/119 (24.37%)	60/119 (50.42%)	89/119 (74.78%)	
PLA2R +	
low titer(20–180 RU/ml)(N = 76)	22/76(28.95%)	40/76(52.63%)	62/76(81.58%)*	
high-titer(>180 RU/mL)(N = 43)	7/43(16.28%)	20/43(46.51%)	27/43(62.79%)	
Pred+Ctx	
PLA2R − (N = 15)	9/15 (60%)	4/15 (26.67%)	13/15 (86.67%)**	
PLA2R + (N = 55)	16/55 (29.09%)	24/55 (45.45%)	41/55 (74.54%)	
Ped+Csa/ F506	
PLA2R − (N = 25)	11/25 (44%)	13/25 (54.16%)	24/25 (96%)	
PLA2R + (N = 61)	12/61 (19.67%)	34/61 (55.73)	46/61 (75.41%)	
Ped+Csa	
PLA2R − (N = 9)	6/9 (66.67%)	3/9 (33.33%)	9/9 (100%)*	
PLA2R + (N = 31)	6/31 (19.35%)	16/31 (51.61%)	22/31 (70.97%)	
Ped+F506	
PLA2R − (N = 16)	5/16 (31.25%)	10/16 (62.5%)	15/16 (93.75)	
PLA2R + (N = 30)	6/30 (20%)	18/30 (60%)	24/30 (80%)	
Notes.

CR complete remission

PR partial remission

CsA cyclosporin

Pred prednisone

TAC tacrolimus

CTX cyclophosphamide

MN membranous nephropathy

Compared with the PLa2R− group and the PLA2R+ group, P∗ < 0.05, P∗∗ < 0.01.

Table 4 Comparison of data between the two groups of patients after 1 year of follow-up treatment.

Total	24HUPRO (g/L)	ALB (g/L)	CREA (µmol/L)	BUN (mmol/L)	eGFR	
PLA2R −(N = 43)	0.78 ± 1.10	38.21 ± 6.14	64.18 ± 14.37	5.37 ± 1.50	107.77 ± 24.43	
PLA2R +(N = 119)	2.22 ± 2.10**	35.09 ± 6.97*	71.47 ± 23.91	6.68 ± 2.72*	105.20 ± 28.09	
Pred+CTx	
PLA2R −(N = 15)	0.79 ± 1.24	37.37 ± 6.80	59.54 ± 12.89	4.83 ± 1.23	118.65 ± 27.28	
PLA2R +(N = 55)	2.24 ± 2.02	34.28 ± 6.46	63.01 ± 23.25	5.75 ± 1.68	114.90 ± 26.31	
Ped+Csa	
PLA2R −(N = 9)	0.21 ± 0.25	40.01 ± 4.49	72.36 ± 14.42	6.25 ± 2.20	92.84 ± 17.14	
PLA2R +(N = 31)	2.54 ± 2.45**	35.02 ± 7.41	80.85 ± 23.66	7.41 ± 2.62	93.74 ± 23.15	
Ped+F506	
PLA2R −(N = 16)	1.18 ± 0.61	38.18 ± 5.94	67.77 ± 15.70	5.59 ± 1.55	115.02 ± 28.47	
PLA2R +(N = 30)	1.80 ± 1.77	36.66 ± 7.08	73.19 ± 22.95	7.08 ± 3.75	114.75 ± 33.04	
Notes.

eGFR: estimated glomerular filtration rate Compared with the PLa2R− group and the PLA2R+ group, P∗ < 0.05, P∗∗ < 0.01.

Discussion

IMN is an autoimmune disease caused by endogenous antigens combined with specific antibodies in the circulation to form in situ immune complexes that are deposited in renal tissues. The anti-PLA2R antibody in circulation binds to PLA2R on the surface of podocytes to form an electron-dense deposit under the foot process and epithelium, thereby activating the complement-forming membrane attack complex, causing podocyte and kidney damage (Glassock, 2012). Several studies have demonstrated that the anti-PLA2R antibody is a key factor in the pathogenesis of IMN. However, there are few studies on patients with IMN who are negative for the serum anti-PLA2R antibody, and the pathogenesis and prognosis of patients with IMN who are negative for the anti-PLA2R antibody are not very clear. In this study, we first studied the clinicopathological features and prognosis of IMN patients who were seronegative for the anti-PLA2R antibody.

Herein, 74.23% of the 229 patients with IMN were positive for the antibody, which was consistent with previous studies, as the rate generally ranges from 71.0–77.8% in IMN patients (Kanigicherla et al., 2013). The 24-hour urine protein, BMG, and creatinine levels were significantly lower in the negative group than in the positive group, and plasma albumin levels were higher in the negative group than in the positive group. Other clinical studies also showed that patients in the PLA2R− group had lower 24-hour urine protein and higher albumin levels than those in the antibody-positive group, which is basically consistent with our findings (Hofstra et al., 2011; Oh et al., 2013; Radice et al., 2016). In this study, the antibody titer in the positive group was positively correlated with the 24-hour urine protein level and negatively correlated with the serum albumin level. Hofstra et al. (2011) also showed that the anti-PLA2R antibody level was positively correlated with the urine protein level and negatively correlated with the albumin level. However, their study also showed a positive correlation between antibodies and serum creatinine, which was inconsistent with the results of our study. The reason for this discrepancy may be related to the facts that the renal function of the patients in our study was within the normal range and that the clinical symptoms were relatively mild. In terms of renal pathology, the incidence of acute and chronic tubular injury was significantly lower in the negative group than in the positive group, which was consistent with the results of Hihara et al. (2016) and indicates that the pathological damage of patients in the antibody-negative group is less severe than that of patients in the positive group.

Figure 1 Detection of PLA2R antigen and IgG4 in renal tissue.

The expression of PLA2R and IgG4 in glomeruli was observed by immunofluorescence microscopy in patients with negative anti-PLA2R antibodies. Simultaneous staining of PLA2R and IgG4 is positive were shown in A and B; simultaneous staining of PLA2R and IgG4 is negative were shown in C and D; E and F show that PLA2R staining is negative and igG4 staining is positive.

Table 5 Comparison of prognosis between the two groups.

Prognosis	PLA2R −(N = 43)	PLA2R +(N = 119)	
Kidney death	0	5	
Clinical death	0	1	
Severe infection	0	3	
Tumor	0	1	
Complete remission	31	14	
Partial remission	21	60	

Figure 2 The log-rank test showed no significant difference in renal survival between the two groups.

Table 6 Factors related to kidney death in patients with IMN (Cox single factor analysis).

							95% Exp(B)’Cl	
	B	SE	Ward	Degrees of freedom	Saliency	Exp (B)	Lower limit	Upper limit	
24HUPRO	.236	.096	6.100	1	.014	1.266	1.050	1.527	
β2-MG	.651	.249	6.822	1	.009	1.917	1.176	3.123	
CRTL	−1.880	.817	5.298	1	.021	.153	1.322	32.516	
ARTL	−2.169	.818	7.038	1	.008	1.114	1.762	43.468	
Notes.

CRTL chronic renal tubular lesions

ARTL acute renal tubular lesions

β2-MG β2-microglobulin

24HUPRO 24-hour urine

Table 7 Multivariate Cox regression analysis of renal death risk in patients with IMN.

							95% Exp(B)’Cl	
	B	SE	Ward	Degrees of freedom	Saliency	Exp (B)	Lower limit	Upper limit	
24HUPRO	.291	.138	4.483	1	.034	1.338	1.022	1.752	
β2-MG	.971	.532	3.334	1	.068	2.641	.931	7.491	
CRTL	−1.937	1.607	1.453	1	.228	.144	.006	3.360	
ARTL	−.668	1.062	.395	1	.530	1.949	.243	15.636	
Notes.

CRTL chronic renal tubular lesions

ARTL acute renal tubular lesions

β2-MG β2-microglobulin

24HUPRO 24-hour urine

In our study, we selected patients with an average daily urine protein level ≥ 6 g, and these patients had at least a moderate or high risk of progressive disease. In addition, CR can lead to a good long-term prognosis, and PR can independently reduce the risk of renal failure (Troyanov et al., 2004). Therefore, early therapeutic intervention may be a more effective approach for treating IMN patients with severe proteinuria. For newly diagnosed patients, on the basis of adjuvant therapies such as antihypertensive, lipid-lowering and anticoagulative treatments, we preferred hormone therapy with CTX or a semiquantitative hormone plus tacrolimus (FK506) and CSA. Herein, the remission rate of the negative group was significantly higher than that of the positive group, especially the CR rate. To the best of our knowledge, this study is the first to report that patients in the negative group had a better clinical remission rate than those in the positive group. Furthermore, in the positive group, patients with a low titer had low 24-hour proteinuria levels, high albumin levels, mild pathological damage, and higher remission rates than those with a high antibody titer, thus suggesting that the patients with low antibody titers were more likely to achieve clinical remission. Rodas et al. (2019) found that spontaneous CR did not occur in patients with PLA2R levels >40 IU/mL and was less frequently observed in patients with a proteinuria level >8 g/day. Furthermore, patients with high antibody levels had a higher risk of developing ESRD. The authors also found that patients with low anti-PLA2R antibody titers (especially antibody titers <40 UI/mL) and proteinuria levels <4 g/day had high spontaneous remission rates. These studies are consistent with our findings, indicating that the anti-PLA2R antibodies titer is strongly correlated with the disease activity of IMN.

Survival analysis in this study showed that the survival rate of the PLA2R− group was higher than that of the PLA2R+ group, but this difference was not statistically significant. Risk factor analysis results showed that BMG, 24-hour urine protein, acute and chronic tubular disease were risk factors, but only 24-hour urine protein was an independent risk factor for renal death. In a predictive model of IMN progression risk, Cattran et al. (1997) and Pei, Cattran & Greenwood (1992) proposed that patients with abnormal or worsening serum creatinine or proteinuria, especially those with a duration exceeding 6 months, had a 72% chance of developing end-stage renal failure. Ponticelli et al. (1989) and Shiiki et al. (2004) also showed that high serum creatinine concentrations, severe proteinuria, and chronic tubulointerstitial lesions were important predictors of the risk of IMN progressing to ESRD. Various clinical indicators, such as severe proteinuria, decreased renal function at diagnosis, and development of renal tubular interstitial lesions, can be considered risk factors for the progression of IMN to renal failure.

Herein, 43 of 162 IMN patients were negative for the serum anti-PLA2R antibody, potentially due to the following reasons: 1. The serum anti-PLA2R antibody was negative, but the PLA2R antigen in kidney tissue was positive; 2. other pathogenic antigens and specific autoantibodies may have existed in the patients with IMN; 3. other secondary factors that have not yet been discovered played roles; 4. the anti-PLA2R antibody in the sera of patients with IMN was indeed negative; 5. the patients’ conditions were in a stable state, and there was no immune activity in the body; and 6. the limitations of the detection technology itself. Therefore, we stained for the PLA2R antigen in the kidney tissues of 43 patients who were negative for the serum anti-PLA2R antibody, and the granular staining rate for the PLA2R antigen in kidney tissue was 62.79%. Consequently, for patients with IMN who were negative for serum anti-PLA2R antibodies, we could routinely perform renal tissue PLA2R antigen staining. This result is also consistent with the findings of Hill, McRae & Dwyer (2016), who found that the PLA2R antigen was potentially positive in renal tissue when serum antibodies were absent in IMN patients, and when the serum anti-PLA2R antibody was detected in combination with the PLA2R antigen in renal tissue, the sensitivity of disease diagnosis could be increased to 95.2%. Detection of the PLA2R antigen in renal tissue is more sensitive and specific for the diagnosis of IMN and can be used as a supplementary detection method for patients who have undergone renal biopsy and are negative for the serum anti-PLA2R antibody (Hofstra & Wetzels, 2014; Svobodova et al., 2013).

For patients with IMN who were negative for the anti-PLA2R antibody, we also considered the presence of other antibodies with structural functions similar to those of the anti-PLA2R antibody. Several clinical reports on other pathogenic antigens of IMN, including thrombospondin type-1 domain-containing 7A (THSD7A) (Tomas et al., 2014), superoxide dismutase (SOD2), α-enolase (a-ENO), aldose reductase (AR) (Bruschi et al., 2011; Prunotto et al., 2010), neutral endopeptidase (NEP), bovine serum albumin (BSA) and BSA antibodies (Murtas et al., 2012), have been published. In 2014, Tomas et al. found THSD7A in the sera of patients with IMN who were negative for the PLA2R antibody. This study suggested that antibodies specific for this antigen were detectable only in the sera of anti-PLA2R antibody-negative IMN patients and were almost undetectable in patients with SMN and other glomerular diseases. The anti-THSD7A antibody may become a highly specific new indicator of disease diagnosis and activity monitoring in IMN patients who are negative for the serum anti-PLA2R antibody (Tomas et al., 2014). In addition, some studies showed that the serum anti-PLA2R antibody, anti-THSD7A antibody, anti-SOD2 antibody, and anti-ENO antibody were all negative in IMN patients with a low proportion of nephrotic syndrome at onset, high plasma albumin level, and high disease remission rate (Murtas et al., 2012). Therefore, the anti-THSD7A antibody, anti-SOD2 antibody, and anti-α ENO antibody in the sera of patients with IMN who are negative for the anti-PLA2R antibody can potentially indicate that the patient’s clinical severity is less severe, and clinical remission can be achieved more quickly. We also need to conduct a larger number of clinical data studies in the future to confirm these results. In this study, we did not find new tumors in the antibody-negative group, and a benign lung tumor occurred in one patient in the antibody-positive group. This finding was not exactly the same as the relevant research results. Previous clinical studies showed that patients with IMN were negative for the anti-PLA2R antibody, especially when they had antibodies against THSD7A. Furthermore, 21% of patients with IMN associated with THSD7A were found to have tumors within 3 months of being diagnosed with membranous nephropathy (Hoxha et al., 2016). Qin et al. (2011) detected serum anti-PLA2R autoantibodies in 10 patients with tumor-associated membranous nephropathy. Seven of the 10 patients were negative for the anti-PLA2R antibody, while only 3 patients were positive, and most patients with tumor-associated membranous nephropathy were considered to be negative for the antibody in the study. In the antibody-negative group in our study, there were no tumors (low tumor incidence rate), the clinical remission rate was high, and the prognosis was good. The reasons for our results may be related to the short follow-up time, loss of follow-up, and other pathogenic factors. We also need to further follow-up with the negative group to observe the relationship between tumors and the anti-PLA2R antibody.

Immunosuppressive therapy has been well proven in patients with IMN. However, because this treatment regimen may cause potential side effects, it must be reserved for patients who truly benefit from this treatment. This retrospective study also found that patients with IMN who were negative for the anti-PLA2R antibody had fewer clinical manifestations and renal pathological changes, and numerous side effects of early immunosuppressive therapy were observed in these patients. Therefore, whether patients with IMN who are negative for the anti-PLA2R antibody and without tumors and other secondary factors can actively use immunosuppressive therapy at an early stage to achieve disease remission will be a new direction of our future research. In addition, for patients with IMN who are negative for the anti-PLA2R antibody, if other specific antibodies are not detected in their serum and the renal tissue PLA2R antigen staining result is also negative, we should consider whether other secondary findings exist that have not yet been discovered, such as tumors, heavy metals, and metabolic factors.

Conclusions

This study focused on the clinical significance of negativity for the serum anti-PLA2R antibody in IMN. The patients who were negative for the serum anti-PLA2R antibody had mild clinical manifestations, mild pathological damage, and relatively high clinical remission rates. Renal tissue PLA2R antigen testing can be considered for patients with seronegative IMN to increase the diagnostic rate of IMN. In addition, studies have shown that 24-hour urine protein is an independent risk factor for kidney death. This retrospective study has some limitations, as the study sample size was limited, and the follow-up time was relatively short. The long-term prognosis and factors related to survival analysis need to be further studied.

Supplemental Information

File S1 Clinical data of 229 cases of membranous nephropathy

Click here for additional data file.

File S2 Clinical data of 162 patients with membranous nephropathy who were followed up

Click here for additional data file.

Supplemental Information 3 The significance of categorical data recorded in digital form in Table1 and Table 2

Click here for additional data file.

We thank Yongmei Wang, Shiyin Jiang, Ruihua Song and Shimin Zhao for their help with data collection.

Additional Information and Declarations

Competing Interests

Author Contributions

Human Ethics

Data Availability

The authors declare there are no competing interests.

Wenkai Guo and Yan Zhang conceived and designed the experiments, performed the experiments, analyzed the data, prepared figures and/or tables, authored or reviewed drafts of the paper, and approved the final draft.

Caifeng Gao and Jiatong Li performed the experiments, prepared figures and/or tables, and approved the final draft.

Jing Huang performed the experiments, authored or reviewed drafts of the paper, and approved the final draft.

Rong Wang analyzed the data, authored or reviewed drafts of the paper, and approved the final draft.

Bing Chen conceived and designed the experiments, performed the experiments, analyzed the data, prepared figures and/or tables, authored or reviewed drafts of the paper, and approved the final draft.

The following information was supplied relating to ethical approvals (i.e., approving body and any reference numbers):

Ethics approval, consent to participate, and data collection of related cases has been approved by the Ethics Committee of Shandong Provincial Hospital Affiliated to Shandong University (LCYJ:No.2019-105).

The following information was supplied regarding data availability:

The raw measurements are available in the Supplemental Files.

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
