# Peer review of "Retrospective study: clinicopathological features and prognosis of idiopathic membranous nephropathy with seronegative anti-phospholipase A2 receptor antibody"

_PeerJ, doi:10.7717/peerj.8650_

## Round 0.1 · original submission · Major Revisions

The introduction can be rephrased for the PLA2 pathology.

Please address all reviewer concerns

Reviewer 1 ·

Basic reporting

Below some suggestions:

- Overall, the article includes sufficient background and the role of anti-PLA2R antibodies is extensively debated and appropriately referenced. . However, the Introduction fails to discuss the rational of the PLA2R antigen assessment. Then, the Authors should also remark on this topic into this section.

- The current phrasing is sometime redundant and the general structure of the article should be revised. In brief, to improve the readability of the text, the Authors should simplify and rewrite some sentences. For instance,

- Line 76 and 77: “titer level” could be replaced with “titer” or “level”.
- Line from 85 to 91: the brief description of the study aim and design should include only general information to make the message clearer. Let me give you an example on how to rewrite this paragraph: “In this study, we performed a two-step evaluation: firstly, we retrospectively compared clinical and pathological features of anti-PLA2R negative (PLA2R-) with anti-PLA2R positive (PLA2R+) subjects; then, we evaluated the prognostic value of PLA2R negative result using a longitudinal design and analysis.”
- Line 105: the clinical and laboratory data should be organized in a more logical way. For example, “General demographic (age, gender) and clinical (duration of disease, systolic and diastolic blood pressure, etc.) records were collected. Laboratory data included general tests (…) and specific biomarkers of renal function and disease activity (e.g. urinary protein excretion (g/day), serum creatinine (mg/dL), etc…)”.
- Line 112: antiphospholipid enzyme A2 receptor (PLA2R) antibody should be removed as this biomarker is extensively described in the next paragraph.
- Line 117: the sentence “The eGFR was calculated based on the MDRD formula” should move in the previous paragraph in the part of the text where the specific biomarkers are listed.
- Line 151: Statistical method paragraph can be simplified and also needs some language revision; line 152: M(1/4,3/4) should be replaced with median and 25°/75° percentiles.

Furthermore, below some typing error to be revised:
- Line 110: "macroglobulin" should be replaced with "microglobulin".
- Line 165: “Result” should be replaced with “Results”.
- Line 312: “PRUE A HILL” should be replaced with “Hill”.
- Line 373: “study” should be replaced with “studies”.
- Line 402: “REFERENCE” should be replaced with “REFERENCES”.

Furthermore, PeerJ standards do not require numbered subsections and the titles of the paragraphs in Materials and Methods should be included or replaced with more appropriate ones to make the message straightforward. On the other hands, the subsections in Results sound redundant; my suggestion is to write the results in a single paragraph.
For instance (these are just suggestions):
- Line 103: “Research method” is redundant.
- Line 104: “Clinical and laboratory data collection”
- Line 114: “Anti-PLA2R detection”
- Line 119: “PLA2R antigen detection”
- Line 124: “Renal biopsy”
- Line 142: “Outcomes in followed up patients”

Experimental design

- The description of the methods needs some minor revision. In particular, the paragraph including the description of the study subjects (line 93) should also include the diagnostic criteria for iMN with some more detail in renal biopsy characteristics. This description should be included before the exclusion criteria (line 96).

- Statistical method (line 154): the Authors stated that they used SNK test to compare two groups. Are they sure? This test is usually performed for comparing three or more sample means. Furthermore, logrank test should be included in this paragraph.

Validity of the findings

- The Authors should clarify the total number of the enrolled patients. In fact, at line 166 they stated that 229 patients with iMN were retrospectively collected, including 59 and 119 PLA2R negative and positive patients, respectively. However, 59 plus 119 equals to 178. Then, what about the remaining 51 subjects?

- The statistical results that are included in parentheses should be integrated including the value of statistical test, the degree of freedom when requested and the right value of p-value (not p<0.05). For example, “(chi2=…, df=…, p=…)” or (KW=…, p=…). This is for line 170,172,175, etc.

- In tables 6 and 7, confidence intevals of B coefficients or of Exp(B) should be reported.

Reviewer 2 ·

Basic reporting

no comment

Experimental design

no comment

Validity of the findings

no comment

Additional comments

Dr. Guo et al. measured serum PLA2R Ab by using EUROIMMUN ELISA as well as staining for PLA2R Ag and IgG4. They found that seronegative for PLA2R Ab IMN patients showed milder disease and better renal prognosis. They claimed that renal pathological staining for PLA2R is important for IMN patients especially when serum PLA2R Ab was negative. The findings are clinically important. However, I still have several concerns.

1. Title
Clinicopathological features and prognosis of idiopathic membranous nephropathy with
negative anti-phospholipase A2 receptor antibody

This title may mislead the readers. To be precise, the title should be modified as follows:
“Clinicopathological features and prognosis of idiopathic membranous nephropathy with
seronegative anti-phospholipase A2 receptor antibody”


2. Lines 40~41 in ABSTRACT:
Otherwise, the PLA2R antigen-positive rate of 43 patients in the PLA2R-group was 62.79%.

I do not understand what “Otherwise” means in this sentence.
Moreover, the PLA2R antigen-positive should exist at least in low level in normal kidney podocyte as shown by Beck et al. (N Engl J Med. 2009 Jul 2;361(1):11-21. doi: 10.1056/NEJMoa0810457.) Therefore, “the PLA2R antigen-positive rate” should be replaced by some other phrases such as “the PLA2R antigen-positive staining rate,” or “the rate of granular staining for PLA2R antigen.

3. Line 301, DISCUSSION
This study showed that 43 of 162 patients with IMN were negative for anti-PLA2R antibodies,

This sentence should be modified as follows:
This study showed that 43 of 162 patients with IMN were negative for anti-PLA2R antibodies in the sera,

Authors should be more carful in describing these critical points.

4. Line 303, DISCUSSION
The anti-PLA2R antibody in the serum is negative, but PLA2R antigen in the kidney
tissue is positive.

Again, I do not agree with this sentence, because PLA2R antigen in the kidney tissue should always be positive at least in a low level. This sentence should be, “The anti-PLA2R antibody in the serum is negative, but PLA2R antibody is positive in the kidney tissue.”

5. Figure 2
Pictures of immunohistochemistry is not very convincing. The authors should show better photos with stronger stanning. Pictures with higher magnifications are also needed.

Reviewer 3 ·

Basic reporting

The paper is of interest however the literature reference should include all the papers about the topic.
There are few typing errors, the English style should be revised.

Experimental design

The authors should refer to the shared definition of complete remission.

It is not clear when the PLA2R-antibodies have been tested, the authors hypothesized that in the negative group ' The patient's condition is in a stable state, and there is currently no immune activity in the body'. Are all the PLA2r tests performed at the time of the biopsy?

Since the authors are defining the complete remission as urine protein<0,3g/day and normal kidney function, the difference in the treatment group (more patients are treated with cyclosporine in the positive group) and the chronic pathological lesions at the diagnosis could be a bias.

Validity of the findings

the design of the study is influencing the validity of the findings

Additional comments

The paper is of interest and it is analyzing the difference among the PLA2R positive and negative iMN subgroups. However the differences in the baseline characteristics could had influenced the outcomes.

---

## Round 0.2 · Minor Revisions

Please carefully check your revision and amend according to Rev 1

Reviewer 1 ·

Basic reporting

Please, add some minor revision to statistical methods. Actually, the term "abnormal distribution" is unusual. Furthermore, I disagree with this sentence “and a P value <0.01 was considered notably statistically significant”. Indeed, in statistics the strength of an association is stated by odds ratio or risk ratio, etc... Conversely, a lower p-value does not mean a greater probability of the association.
Below some suggestions:

Descriptive statistics were used to calculate mean and standard deviation for quantitative variables with normal distributions, median and percentiles for non-normal distributions and percentage for categorical variables. The comparisons between independent groups were performed by t-test, non-parametric tests and Chi-squared for continuous or categorical variable, respectively. Correlation between several parameters was expressed as Spearman rank coefficient of correlation. Patient and renal survival probabilities were estimated by the Kaplan-Meier method. The relationships of the covariates to patient and renal survival were evaluated by univariate analysis with the log-rank test and by multivariate analysis with the Cox proportional hazards model; the Harrel C statistic was used for verification of the risk factors confirmed by the Cox proportional hazards model. A Cox analysis of risk factors was performed in combination with clinical significance. A p-value < 0.05 was considered statistically significant.
Statistical analyses were performed using IBM SPSS Statistics for Windows, version 19.0 (IBM Corp., Armonk, N.Y., USA).

Experimental design

I agree with the revised version

Validity of the findings

I agree with the revised version

Additional comments

I thank the Authors for their effort in revising the manuscript. The current version improved a lot the structure and clarity of the manuscript. I finally ask them to add some minor revision into the statistical method section as suggested above.

---

## Round 0.3 · Minor Revisions

I believe that the statistics section should be further improved. Please can you work further on this part of the manuscript.

---

## Round 0.4 · accepted · Accept

Congratulations !!!
Your study has been accepted.